Geographic distribution modeling and taxonomy of Stephadiscus lyratus (Cothouny in Gould, 1846) (Charopidae) reveal potential distributional areas of the species along the Patagonian Forests

http://orcid.org/0000-0002-3531-1744 Cuezzo Maria Gabriela gcuezzo@webmail.unt.edu.ar
Medina Regina Gabriela regina.g.medina@gmail.com
Nieto Carolina
Instituto de Biodiversidad Neotropical, Consejo Nacional de Investigaciones Científicas y Técnicas (CONICET)-Universidad Nacional de Tucuman (UNT) , Yerba Buena, Horco Molle, Tucumán , Argentina
Morrone Juan J.
Electronic publication date: 2021 Jul 5
Publication date: 2021
Volume: 9
Electronic Location ID: e11614
Received 2020 Dec 23; Accepted 2021 May 24
Copyright: © 2021 Cuezzo et al.
Copyright year: 2021
Copyright holder: Cuezzo et al.
License: This is an open access article distributed under the terms of the Creative Commons Attribution License, which permits unrestricted use, distribution, reproduction and adaptation in any medium and for any purpose provided that it is properly attributed. For attribution, the original author(s), title, publication source (PeerJ) and either DOI or URL of the article must be cited.
License URL: https://creativecommons.org/licenses/by/4.0/

Keywords: Stylommatophora, Tierra del Fuego, Ecological niche modeling, Land cover, Protected area, Microhabitat preference

Funding: Argentine National Council of Scientific Research (CONICET) PIP 0050 and P-UE 0099 The financial support to carry out this research was provided by the Argentine National Council of Scientific Research (CONICET) through the grants PIP 0050 and P-UE 0099. The funders had no role in study design, data collection and analysis, decision to publish, or preparation of the manuscript.

==============================
Background

Stephadiscus lyratus (Couthouy in Gould, 1846), an endemic Charopidae from southern South America, was described from few dry shells. The distribution of this species is known on scattering occurrences, mainly from material deposited in museum collections. We provide here new information on anatomy, habitat, and microhabitat preferences and estimate the potential geographic distribution of the species to test if it is exclusively endemic to the Subpolar Magellanic Forest.

Methods

Fieldwork was carried out in the National Parks of the Patagonian Forests. Snails were photographed, measured, and dissected for anatomical studies; shells were studied with scanning electron microscopy. Estimation of the species geographical distribution (EGD) was obtained through correlative ecological niche modeling (ENM). We designed a calibration area a priori with known species points of occurrence in the Magellanic Subpolar Forests and borders of the Patagonian steppe. Seven bioclimatic variables of the WorldClim database were used. The best ENMs were calibrated and selected using a maximum entropy method with Maxent v3.3.3K through the R package “kuenm”. Candidate models were created by combining four values of regularization multiplier and all possible combinations of three feature classes. We evaluated candidate model performance based on significance (partial ROC), omission rates (E = 5%), and model complexity (AICc). From the best models obtained, a final model was transferred to a region “G” consisting of the calibration area plus the Valdivian Temperate Forests and whole Patagonian steppe, where we hypothesize that the species could be present. Finally, we obtained binary presence-absence maps. We quantified the proportion of the occurrence points and distribution range of S. lyratus in different land cover categories. To explore the degree of protection of S. lyratus’EGD, we quantified the proportion of its distributional range within protected areas.

Results

A be-lobed kidney, a close secondary ureter, the terminal portion of the uterus forming a compact glandular mass, and the vas deferens with a dilatation are new anatomical information that distinguishes this species. Stephadiscus lyratus inhabit cold native forest areas, mainly living on or under humid logs in contact with the ground. The main constraining variables to explain S. lyratus distribution in the EGD were BIO3, BIO12, BIO6, and BIO4. The potential area of distribution obtained almost duplicates their original range (140,454 km2) extending to the Valdivian Temperate forests mainly in Chile. Natural and semi-natural terrestrial vegetation was predominant in the potential area of distribution of S. lyratus. However, only 14.7% of this area occurs within current protected areas from Argentina and Chile. The ectothermic physiological traits of this species, low dispersal capacity, and its narrow habitat requirements turn S. lyratus into a potentially vulnerable species.

Introduction

Charopidae is a family of Punctoidea land gastropods with an extensive distribution that includes South America, South Africa, Australia, New Zealand, and Oceania (Salvador et al., 2020). Southern Argentina and Chile are particularly rich in endemic species of Charopidae (Miquel & Cádiz Lorca, 2008). Hylton Scott (1964, 1968, 1970, 1973, 1981) was the researcher that most studied and described Charopidae species in South America. Unfortunately, most of her descriptions were based on a single or few dry shells, which resulted in the absence of intraspecific shell variability studies. The lack of species anatomical data constitutes a barrier for intra family taxonomy. Also, adequate fieldwork to estimate the current distributional range of Charopidae genera and species in South America has not been done.

Stephadiscus Hylton Scott, 1981 originally included some species that are currently classified in Stephanoda Albers, 1860, and Stephacharopa Miquel & Araya, 2013 (Miquel & Barker, 2009; Miquel & Araya, 2013). At present, the genus is formed by six species, Stephadiscus lyratus [designated as the genus type species], S. celinae (Hylton Scott, 1969), S. mirabilis (Hylton Scott, 1968), S. perversus (Hylton Scott, 1969), S. rumbolli (Hylton Scott, 1973), and S. stuardoi Miquel & Barker, 2009. The taxonomic position of Stephadiscus striatus Hylton Scott, 1981 from northeastern Argentina and Venezuela, should have to be reconsidered, as it appears to belong to Punctidae (Miquel & Barker, 2009). Known Stephadiscus distribution is restricted to Patagonia at both sides of the Andes from S 36° towards the southernmost portion of the continent, including Malvinas islands (=Falklands) and southern archipelagos (Miquel & Barker, 2009; Miquel & Araya, 2013) and belong to the Andean region (Morrone, 2018). This biogeographic area has a closer relationship to the Austral region in the Austral kingdom (Morrone, 2015, 2018). The current distributional range of Stephadiscus lyratus (Couthouy in Gould, 1846) had been established on scattering points of occurrences taken from its original description, and subsequent species mentions (Gould, 1845; Hylton Scott, 1972, 1981; Miquel & Araya, 2013, see “Species remarks” section). Erroneous taxonomic identified material from Museum Collections suggested the presence of this species in the forests of northern Patagonia, even though Hylton Scott (1981) stated that S. lyratus could be a strictly Magellanic species. While Stephadiscus celinae, S. perversus, and S. mirabilis also occur in the Valdivian rainforest sub-ecoregion of northern Patagonia, S. rumbolli is exclusively from the southern sub-ecoregions. For this reason, there are doubts as to whether the distribution of S. lyratus extends naturally to the Valdivian rainforest or is restricted to southern areas.

Modern methodologies to estimate a species distributional area involve the ecological niche models (ENMs) that relate the species distribution data (species occurrence at known locations) with information about the environment (abiotic factors) (Beltramino et al., 2015; Medina, Ponssa & Aráoz, 2016). The environmental variables of the localities of occurrence are also informative about the species’ potential distribution. Estimates of geographic range obtained by ENM techniques when data are scarce, or when species are rare, have proven to be more successful than those obtained by traditional methods (e.g., minimum convex polygon) (Marcer et al., 2013; Syfert et al., 2014; Pena Jão et al., 2014). These also allow avoiding the potential subjective bias of experts (Fourcade et al., 2014). Furthermore, identifying the combination of environmental conditions in the relevant scenopoetic variables offers the opportunity to discover populations isolated (Wiens & Graham, 2005). On the other hand, modeling species potential areas of distribution may also provide information on the geographic distribution of unknown sister species (Peterson et al., 2011).

Invertebrates are recognized as indicators of human disturbance, due to their low dispersal capacity and their dependence on microhabitats for survival and mating. Particularly, most snails that are not arboreal are dependent on litter from deciduous trees and have higher abundances in multispecies forests with strong broadleaf components (Addison & Barber, 1997; Niemelä, 1997). Taking into account that many natural areas to date are severely fragmented or threatened by human activities, obtaining information on the ecological aspects of these species is very useful for future biological conservation work (Barahona-Segovia et al., 2019). Previous studies in Stephadiscus lyratus provided little data on habitat or microhabitat preferences and therefore future evaluation on its risk of extinction will need ecological information on this species. The objective of this research is to provide new information on the anatomy and ultrastructure of the Stephadiscus lyratus shell, using recently collected material, identifying and describing the microhabitat where it is found, and analyzing the environmental variables that are limiting its distribution with ENMs techniques.

Finally, we estimate its potential geographic distribution to hypothesize if S. lyratus can also be distributed in the Valdivian Temperate Forest or if this species is endemic exclusively to the Magellanic Subpolar Forest.

Material and Methods

Study area (Figs. 1A and 1B)

Fieldwork was carried out in the Patagonian Forests, also known as the temperate forest of southern South America that extends in a narrow strip of land over the Andes Mountain between 35° and 55° south latitude (Armesto, Rozzi & Caspersen, 2001). In Argentina, this region occupies the western zone from the provinces of Neuquén to Tierra del Fuego and islands of the southern Atlantic. The Patagonian Forests are divided into two different areas, the northern Valdivian Temperate and the southern Magellanic Subpolar Forest ecoregions sensu Olson & Dinerstein (1998), Olson et al. (2001) and Morello et al. (2012) or sub-ecoregions sensu Dos Santos et al. (2020). We followed here this last-mentioned classification.

Figure 1 Study area and collection points of Stephadiscus lyratus.

(A) Magellanic Subpolar Forest, Valdivian Temperate Forest and Patagonian steppe sub ecoregions showing historic records of occurrence. (B) Southern portion of Tierra del Fuego with new records of occurrences. (C and D) Aspect of the Magellanic subpolar forest trees, mainly corresponding to the genus Nothofagus, where the studied species was collected.

The Valdivian Temperate Forest (Fig. 1A) covers a narrow area running from 35° to 48° south latitude between Chile and Argentina. Annual precipitation varies between 1,000 mm in the north and more than 6,000 mm per year in the southern part of the sub-ecoregion. This seasonal precipitation decreases significantly on the eastern slope of the Andes in Argentina, where rainfall of less than 200 mm is recorded only 100 km east of the Andean peaks. Maximum annual average temperatures vary between 21 °C and 13 °C in the northern and southern ends of the sub-ecoregion. Minimum annual average temperatures range from 7 °C to 4 °C (https://www.worldwildlife.org/ecoregions/nt0404). Biogeographically, these forests share floristic similarities with other temperate forests in the southern hemisphere located in Australia and New Zealand (McGlone, Lusk & Armesto, 2016). However, there is a high degree of endemism not only in the flora but also in the fauna at the species level.

The southern areas of the Magellanic Subpolar Forests are well represented in Tierra del Fuego, where they occupy the entire south of the province (Figs. 1A and 1B). However, the dominant vegetation is a forest of less species diversity in comparison to the Valdivian rainforest due to the low temperatures and rainfall.

Tierra del Fuego or Fueguia is the archipelago located south of the Strait of Magallanes between 52° 28′S and 55° 03′S; it occupies about 66,000 km² (Fig. 1A). The main island is Isla Grande, with 48,000 km2 representing 70% of the surface of the archipelago, from which 21,263 km2 belongs to Argentina (Frangi et al., 2004). The vegetation of the island is mainly formed by a Patagonian steppe of grasslands and shrubs located at the northern part, and humid deciduous and evergreen forests plus peat bogs located in the center and southern portion. The trees of the genus Nothofagus dominate the forest composition in Tierra del Fuego and constitute the most austral forest in the world as part of the Magellanic Subpolar forest sub-ecoregion. The highest rainfall is recorded in the south of the island, decreasing to the east and center of it. In the south and west of the island, it is very windy, foggy, and humid most of the year with few days without rain, sleet, hail, or snow. The average annual temperature on the island is 5.5 °C to the north and 5.9 °C to the south. Above the mountains, the temperature decreases with elevation, these gradients determine temperatures below zero in the winter months (Frangi et al., 2004). The Big Island of Tierra del Fuego and the Islas de Los Estados were modeled by the erosive action of glaciers that covered large portions of land on several occasions. The glacial topography and the temperate-cold and humid climate that prevail throughout the year favored the formation of the peat bogs that are now part of the Fuegian landscape.

Fieldwork and specimen collections

Fieldwork was carried out in the Magellanic Subpolar Forests of Tierra del Fuego National Park (DRPA 146/2019) and other non-preserved areas in Isla Grande of Tierra del Fuego, Argentina, during December 2018–January 2019 (Figs. 1C and 1D). In the Valdivian Temperate Forests, fieldwork was done within Los Alerces National Park and Puerto Blest in Nahuel Huapi National Park (DRPA 1674, DFyFS1/19) during January 2020. We qualitatively searched for land snails along transects for half an hour in each collecting site. Searching was mainly focused on microhabitats that seem to be most favorable for snails, such as between exposed roots of trees, under the bark of trees, under rocks, or under tree trunks lying on the forest floor in contact with soil. In each collecting site, we recorded altitude and geographic coordinates. We also took samples of 50 × 50 cm quadrats of leaf litter + 2 cm of topsoil from moist microhabitats. Samples were placed in plastic bags and posteriorly sieved through three decreasing mesh widths (3, 1.5, and 0.5 mm) in the laboratory of the Centro Austral de Investigaciones Cientificas (CADIC-CONICET, Tierra del Fuego) under a stereoscopic microscope. All snails collected were photographed alive before relaxing them in water for 24 h, posteriorly fixed in ethanol 96%, and preserved in ethanol 70% for anatomical studies. Several specimens were also fixed directly in ethanol 96%, without relaxation in water, for future molecular studies. All the material collected was deposited in the Malacological Collection of the Instituto de Biodiversidad Neotropical (IBN, CONICET-UNT, Tucumán, Argentina).

Morphology

Ten adult shells were photographed using a Zeiss Stemi 508 with ActionCam and measured using the software ImageJ 1.49 (Schneider, Rasband & Eliceiri, 2012). Shell measurements selected, on dorsal and lateral views, are major shell diameter (DM), shell minor diameter (Dm), shell height (H), apertural height (Hap), and apertural diameter (Dap). The number of shell whorls was calculated following the Kerney & Cameron (1979) methodology. Photographs and shell measurements were carried out at the CADIC in Tierra del Fuego. For anatomical information, dissections of seven adult specimens were studied under a Leica MZ6 stereoscope. Illustrations of the dissected organs/systems were carried out with the aid of a camera lucida. The terms proximal and distal refer to the position of an organ or part of an organ in relation to the gamete flow from ovotestis (proximal) to genital pore (distal) as in previous works (Cuezzo, 2006; Cuezzo et al., 2018). The distinction of the limits between the epiphallus and penis is based on the internal sculpture of their inner wall. Shell ultrastructure was studied and described using a SEM Zeiss Supra 55VP at the Integral Center of Electron Microscopy (CIME) of the National University of Tucumán, Argentina (UNT).

Occurrence records

We compiled a total of 60 geographic records of Stephadiscus lyratus from field surveys, museum collections, and scientific articles (Fig. 1, Table S1). Twenty new reliable records were obtained through fieldwork in Tierra del Fuego. To avoid primary taxonomic sources of error from specimens deposited at different Museums, we corroborated the taxonomic identification of specimens according to their shell morphology. The malacological collections of IBN (Instituto de Biodiversidad Neotropical, Tucumán, Argentina), MACN-In (Museo Argentino de Ciencias Naturales Bernardino Rivadavia, Buenos Aires, Argentina), and MLP (Museo de La Plata, Buenos Aires, Argentina) were revised accordingly. Other sources of information were electronic databases from the Smithsonian National Museum of Natural History (NMNH), Museum of Comparative Zoology (MCZ), and Academy of Natural Sciences in Philadelphia (ANSP). We used Google Earth to georeferenced localities of occurrences that lacked geographic coordinates. From the total number of records, we removed duplicate records, which left 37 unique sites with trustable geographic information. To avoid over-representation of certain environmental combinations, we spatially filtered records based on a 5 km radius, which left 24 localities, then we split them in 30% for test and 70% for train the ecological niche model, both functions implemented in R-package “ellipsenm” (Cobos et al., 2020).

Estimations of potential geographic distributions (EGDs)

Estimates of the geographical distribution (EGD) of S. lyratus were obtained through correlative ecological niche modeling (ENM). To adequately model the species niche, we used the theoretical Biotic-Abiotic-Mobility framework (Soberón & Peterson, 2005). Only abiotic and mobility factors were taken into account because the biotic components (i.e., biotic interactions) are virtually impossible to spatially quantify thoroughly at regional scales (Peterson et al., 2011). As for the mobility component, we designed a calibration area a priori “M” (Barve et al., 2011) considering sub-ecoregions with known species points of occurrence, i.e., in the Magellanic Subpolar Forest and the southern portion of the Patagonian steppe. To delineate abiotic components, we used 15 bioclimatic variables of the WorldClim database (Hijmans et al., 2005) at a spatial resolution of 30 arc seconds (~1 km2), excluding the four variables that combine temperature and precipitation owing to be known artifacts (Escobar et al., 2014). We clipped the environmental data layers to the calibration area defined. To eliminate one variable per pair of highly correlated variables (r ≥ 0.85), we performed a correlation analysis through the “ntbox” package (Osorio-Olvera et al., 2020). Thus, seven bioclimatic variables were selected for the analyses: BIO1 = Annual mean temperature, BIO2 = Mean diurnal range (Mean of monthly (max temp–min temp)), BIO3 = Isothermality (BIO2/BIO7) (* 100), BIO4 = Temperature seasonality (standard deviation * 100), BIO6 = Min temperature of coldest month, BIO12 = Annual precipitation, and BIO15 = Precipitation seasonality (Coefficient of variation).

The best ENMs were calibrated and selected using a maximum entropy method with Maxent v3.3.3K (Phillips, Anderson & Schapire, 2006) through the R package “kuenm” (Cobos et al., 2019). Then, candidate models were created by combining four values of regularization multiplier (0.1, 0.5, 1, 2), and all possible combinations of three feature classes (linear = l, quadratic = q, product = p). We evaluated candidate model performance based on significance (partial ROC, with 100 iterations and 50 percent of data for bootstrapping), omission rates (E = 5%), and model complexity (AICc). Best models were selected according to the following criteria: (1) significant models with (2) omission rates ≤5%. From among this set, models with delta AICc values of ≤2 were chosen as final models. The final model was created using the spatially filtered records (24 occurrences) and the selected parameterizations. We produced 100 iterations with five replicates by bootstrap, with logistic outputs. We ran the models with no extrapolation or clamping to avoid artificial projections based on extreme values of the bioclimatic variables (Elith et al., 2011; Owens et al., 2013; Merow et al., 2014; Guevara et al., 2018). Then, the final model was transferred to a region “G” consisting of the calibration area plus the Valdivian Temperate Forest and Patagonian steppe regions, where the species could be also present as its congeners. To identify extrapolation risk areas in model transfers, we performed MOP analyses (Owens et al., 2013). This function calculates multivariate environmental distances between sites across the transfer region (G) and the nearest portion of the calibration region to identify areas that have a condition of strict or combinational extrapolation.

Finally, to obtain binary presence-absence maps, we used a minimum presence threshold, modified to consider presence data errors (Peterson, Papeş & Eaton, 2007) implemented in R-package “ENMGadgets” (Barve & Barve, 2019). This modified threshold included 100% of the presence points minus the dataset error (E) (Phillips & Dudík, 2008); we assumed E = 5% based on our experience of obtaining the presence data. This conservative method minimizes the commission error rate.

Spatial analysis

To spatially characterize the distribution of Stephadiscus lyratus we quantified the proportion of occurrences points and their potential distribution in different categories of the land cover of the Globcover 2009 dataset (Arino et al., 2012). The 22 land cover categories (e.g., Closed to open (>15%) broadleaved evergreen or semi-deciduous forest (>5 m), Mosaic forest or shrubland (50–70%)/grassland (20–50%)) are according to the UN Land Cover Classification System (LCCS) (Di Gregorio, 2005).

To explore the degree of protection of S. lyratus EGD, we quantified the proportion of its range within protected areas. We take into account the protected areas categories I to VI assigned by the International Union for Conservation of Nature (IUCN) (Dudley, 2008) and the National Parks, even though these are not included in any IUCN category. Shapefiles of the protected areas were obtained from the World Database of Protected Areas (IUCN, UNEP-WCMC, 2020) and http://mapas.parquesnacionales.gob.ar/.

Results

Systematic account

Superfamily Punctoidea Morse, 1864

Family Charopidae Hutton, 1884

Subfamily Charopinae Hutton, 1884

Stephadiscus Hylton Scott, 1981

Type species: Helix lyratus Couthouy in Gould, 1846, by original designation.

Species description (Figs. 2–4)

Figure 2 External morphology of Stephadiscus lyratus shell and live animal.

(A and B) Live animal from Tierra del Fuego National Park showing natural shell coloration and by transparency, the irregular spots of the lung. Note black animal body with lighter basal foot and mantle collar. (C) Dorsal, (D) ventral and (E) lateral views of shell, scale bar = 1 mm (IBN 951). Photo credit: MG Cuezzo.

Figure 3 Shell ultrastructure of Stephadiscus lyratus.

(A) Ultrastructure of protoconch showing the radial disposition of major ribs, scale bar = 100 µm. (B and C) Details of the body whorl sculpture with major ribs separated at regular spaces, and micro radial ribs. Note nodules supporting ribs, scale bar = 10 µm. (D) Deep suture between body whorls and penultimate whorl, scale bar = 10 µm. (E) Shell umbilicus with sculpture, scale bar = 20 µm. Photo credit: MG Cuezzo.

Figure 4 Morphology of pallial and reproductive systems of Stephadiscus lyratus.

(A) Pallial system with pulmonary and pericardial cavity. Note the bilobated kidney overlapping the pericardium. Small whitish concretion are scatter along the pulmonary roof. (B) General shape of reproductive system showing large, rounded albumen gland and spermoviduct divided in two portions, proximal short, cylindrical, continuous to albumen gland, distal expanded in rounded glandular chamber, scale bar = 2 mm. (C) Detail of the uterus chamber in distal spermoviduct, free oviduct cylindrical, short, scale bar = 2 mm. (D) Duct of bursa copulatrix with basal widening, sac rounded. (E) Detail of the penial complex showing penis with a sac-like appendix. The epiphallus reflected over penis, is continuous with a short finger-like flagellum. Note that the vas deference is delated before inserting into epiphallus and that the dilatation is as long as the flagellum, giving appearance of fork ending epiphallus, scale bar = 2 mm. Abbreviations: ag, albumen gland; bc, bursa copulatrix; dbc, bursa copulatrix duct; e, epiphallus; f, flagellum; fo, free oviduct; hd, hermaphroditic duct; kl1, pericardial side kidney lobe; kl2, rectal side kidney lobe; mc, mantle collar; p, penis, pr, penial retractor; r, rectum; s, Spermoviduct; su, secondary ureter; u, uterus; v, vagina; vd, vas deferens.

Stephadiscus lyratus (Couthouy in Gould, 1846)

Helix lyrata Couthouy in Gould, 1846: 167; Gould, 1852: 39.

Patula rigophila Mabille, 1886: 123.

Amphidoxa lirata Pilsbry, 1894: 41.

Amphidoxa (Stephanoda) lyrata Smith, 1905: 339

Stephanoda lyrata Pilsbry, 1911, 518; Hylton Scott, 1972: 67.

Stephadiscus lyratus Hylton Scott, 1981: 124; Miquel & Araya, 2013: 230.

Syntype: MCZ 88297 MCZ: Museum of Comparative Zoology, Harvard University

Type locality: Orange Harbor, Tierra del Fuego [according to Johnson (1964): located at the west side of Nassau Bay].

Morphology (Figs. 2–4)

External body morphology (Figs. 2A and 2B)

Animal black with lighter foot and mantle collar around shell aperture. A deep longitudinal furrow, the pedal groove, runs parallel to the foot edge on each side and above it. Dark irregular spots are seen for transparency through the shell, although some specimens are lighter. Foot short, triangular pointed, not to slightly surpassing the shell diameter.

Shell (Figs. 2C–2E; 3A–3E): Discoidal, 3½ to 4 convex regularly expanded whorls, low spire depressed, not planispiral as apex elevated, fragile (DM = 4,247–5,041 mm; Dm =3,4 44–4,487 mm; H = 2,283–2,432 mm) with brown caramel to light whitish color (Figs. 2A–2E). Protoconch with 35–40 axial delicate, smooth, elevated ribs separated at regular intervals (Fig. 3A). Spaces between protoconch ribs with thinner axial costulae. Protoconch not clearly delimited from the teleoconch, and similarly sculptured. Teleoconch surface with major axial ribs (Figs. 3B–3D), interspace between them of 4–7 µm filled with 5–7 micro radial costulas in between major ribs, nodules at regular intervals supporting ribs, giving the appearance of radial cords (Figs. 3B and 3C). Deep irregular suture (Fig. 3D). Circular aperture, not descendent, with sharp peristome (Hap = 1,861–2,110; Dap = 1,809–1,998). Umbilicus 1/3 or slightly less of body whorl major diameter, with same sculpture as nepionic portion of the shell (Fig. 3E).

Pallial system (Fig. 4A): Pulmonary roof dark with black spots and whitish small granules over the surface. Spots and granules visible through shell. Pulmonary sac short occupying 1/4 of body whorl. Kidney triangular, bilobated, with pericardial side lobe overlapping pericardium. Rectal side kidney lobe bigger than pericardial arm. Principal pulmonary vein short, slender, not branched, smoothly marked. Remaining pulmonary roof smooth with no other veins. Secondary ureter present, close along its length, parallel to rectum. Pallial gland absent.

Jaw and radula: Jaw consists of narrow vertical plates, slightly arched, cream-colored. Radula as described by Hylton Scott (1970).

Reproductive system (Figs. 4B–4E): Albumen gland shapeless, roughly globular, rounded margins. Spermoviduct formed by prostate and uterus fused. Uterus divided into two portions, proximal short, cylindrical, continuous to albumen gland, distally expanded in a rounded glandular chamber (Figs. 4B and 4C). Free oviduct cylindrical, short. Bursa copulatrix sac round, resting over uterus distal portion. Duct of bursa copulatrix basally thickened, progressively decreasing in diameter towards the sac (Fig. 4D). Vagina as long as bursa copulatrix duct, distally widened, featuring three thick, longitudinal pilasters on the interior wall. Vas deferens cylindrical, narrow in diameter, short, running from basal prostate towards peni-oviducal angle, inserting into epiphallus. Penial retractor thin, inserted in penis. Penis cylindrical, thicker than epiphallus, with a sac-like appendix in upper portion (Figs. 4C–4E). Short verge in upper penial chamber, inner penial wall with short pilaster noticeable towards middle penis length. Epiphallus reflected over penis, shorter, thinner ending in a short finger-like flagellum (Fig. 4E). Vas deference delated before inserting into epiphallus anteriorly of penial retractor insertion, dilatation as long as flagellum, giving the appearance of fork ending epiphallus.

Microhabitat characterization

Rainforests, including temperate forests, provide a variety of living spaces where snails can feed, crawl, and live. Although micro snails are usually associated with leaf litter, Stephadiscus lyratus was mainly found living on or under the bark of fallen trees or under humid logs in contact with the ground (Figs. 1C and 1D). We found actively crawling snails only on tree barks or on moss logs. This species is not considered to be arboreal or semi-arboreal. No specimens were found in living trees, nor in their leaves or shrubs, the majority of alive snails were found under fallen decaying logs in contact with the ground. During the hibernation period in wintertime, these microhabitats in contact with soil can act as a buffer and help the species to survive during extreme freezing conditions. Dry shells were recovered from soil samples, but in general, they were worn out. Feeding habits in this species are not known. However, the diversity of fungi from the decaying wood is an important food resource for snails (Solem, 1982; Barker, 2001). Species of the genus Radiodiscus occur in sympatry with S. lyratus. In places outside protected areas where the forest shows some degree of alteration, living snails were scarce or difficult to find.

Species remarks

The history of the species discovery as well as the problem of the species authorship is explained in Supplemental Material (See Appendix).

Comparison with species of the same genus: S. lyratus has the largest shell diameter (DM = 5.5 mm) among all species classified in Stephadiscus. It is a very conspicuous species regarding its shell coloration and sculpture. Although S. perversus is similar in shell coloration, it cannot be confused with S. lyratus because it is sinistrorse and has a smaller shell (DM = 2.8; H = 1.5 mm). Stephadiscus lyratus is also similar in shell sculpture and general shape to S. stuardoi (DM = 2.0–2.02, H = 0.85), but differs from this species in its larger shell diameter and the absence of weak spiral threads in the teleoconch. In the original description, Couthouy in Gould mentioned that S. lyratus could be a synonym of Helix costellata d’Orbigny, 1835 (now Zilchogyra costellata (d’Orbigny, 1835)). However, H. costellata is distributed in the Humid Pampa ecoregion in Buenos Aires, Argentina, an extra Patagonian area with completely different ecological requirements. H. costellata (DM = 4 mm, H = 2 mm) is smaller in shell diameter and height than S. lyratus and its protoconch is a smooth surface.

Stephadiscus lyrata and S. mirabilis are the only species of the genus with known anatomies. Both show a rounded glandular mass, identified as a “dilated sac” by Hylton Scott (1970) at the base of the spermiduct, in the distal genitalia. A similar structure is also present in Stephacharopa testalba (Hylton Scott, 1970). Most of the described species of the genus have been originally established only on single or two dry shells (Table S2), and after their discovery, rarely fresh specimens have been collected in their area of distribution.

Comparison with related genera: Stephadiscus is defined by having plane-convex whorls, presence of a protoconch, and teleoconch with similar ornamentation consisting of thin, nodulose ribs, without a marked limit between the protoconch and the teleoconch (Hylton Scott, 1981; Miquel & Araya, 2013). Therefore, the transition of protoconch towards teleoconch is barely distinguished. These ribs increase in height towards the body whorl, with thinner costula in the interspaces. Stephadiscus is different from Stephacharopa Miquel & Araya, 2013 because in the latter genus, the sculptured protoconch is dissimilar to the teleoconch ornamentation while in the former, the sculptured protoconch is similar to the one present in the teleoconch. Stephadiscus differs from Stephanoda mainly in the sculpture of the protoconch, since the latter possesses a reticulated pattern. Differences in anatomy are not possible to be established for the lack of studies on these genera. Other South American charopid genera, such as Lilloiconcha Weyrauch, 1965 and Zylchogyra Weyrauch, 1965 can reach similar shell sizes but differ from Stephadiscus in general shell shape and in that their protoconch is smooth (Miquel & Araya, 2013). S. lyratus shows a specialized vas deferens-epiphallus junction as other Charopinae from the Pacific Islands described by Solem (1982), marking a difference with subfamilies Semperdoninae, Trukcharopinae, and Rotadiscinae.

Estimations of potential geographic distributions (EGDs)

We obtained 24 candidate models statistically significantly better than null expectations (i.e., predictions from the models coincided with testing occurrence data more frequently than would be expected by random association of points and a prediction of that areal extent) (Table S3). From these, only one final model was selected that was statistically significant and met the AICc criteria (two of three selection criteria) (Table S4). The chosen settings were linear, quadratic, and product features and 0.1 of regularization multiplier. The bioclimatic variables that most contribute to the model were BIO3 = Isothermality (BIO2/BIO7) (×100), BIO12 = Annual precipitation, BIO6 = Minimum temperature of the coldest month, and BIO4 = Temperature seasonality (standard deviation of mean month temperature * 100) (47%, 17%, 12% and 11.5% percent of contribution, respectively). Response curves also gave an indication of the range under which the variable reaches its optimum suitability. The optimum suitability for isothermality is around 43% to 55%, from here decreased abruptly to cero, which indicates that day to night temperature oscillations are smaller than annual temperature fluctuations (Fig. 5A). BIO4, BIO6, and BIO12 displayed a bell-shaped response of increased suitability as the variable increases above a certain value (Figs. 5B–5D). The optimum suitability of BIO4 is around 2,750, this is a measure of temperature change over the year, the larger the value (standard deviation of mean monthly temperature x 100), the greater the variability of temperature (O’Donnell & Ignizio, 2012). The optimum suitability of BIO6 is between −2 and 0 °C (around −1 °C, Fig. 5C), this is a measure of minimum cold temperatures throughout the year. In the case of BIO12, the species had its maximum suitability in the 500–600 mm within a narrow range that abruptly decreased when the precipitation increased above this threshold (Fig. 5D).

Figure 5 Response curves for the most important variables in the Maxent model for the environmental suitability of Stephadiscus lyratus.

(A) BIO3: Isothermality, (B) BIO4: Seasonality, (C) BIO6: Minimum temperature of the coldest month, and (D) BIO12: Annual precipitation. The red lines indicate the mean values, while blue areas denote 1 standard deviation limits, resulting from bootstrap replicates in model runs.

Spatial analysis

The known area of distribution of S. lyratus was approximately 72,672 km2, mainly coincident in the southern portion of the Magellanic Subpolar Forest (below –51° of latitude), at both sides of the Andes and marginally in the southern part of Patagonian steppe in Tierra del Fuego (Fig. 6A). The MOP analysis (Fig. S1) indicated that areas with the most dissimilar variables conditions (i.e., where one or more environmental variables are outside the range present in the training data) were found beyond the potential distributional areas predicted by the model in the “G” area.

Figure 6 Estimated and Potential areas of occurrence of Stephadiscus lyratus.

(A) Valdivian Temperate Forest and Magellanic Subpolar Forest sub ecoregions, showing “M” area and the estimated geographic distribution (EGD). (B) Final model transferred to region “G”, with the potential area of distribution of Stephadiscus lyratus. Note that areas of the model coincident with the Valdivian Temperate Forest mostly occur in the Chilean portion of the sub ecoregion.

When the final model is transferred to region “G”, we found that the potential area of distribution almost duplicates their original range (140,454 Km2). This new region extends mainly to the Valdivian Temperate Forest between –40 and –46 latitude, mostly in Chile and a small portion of Argentina, while towards the Patagonian steppe increase marginally (Fig. 6B).

Natural and semi-natural terrestrial vegetation was predominant in the occurrence points and the potential area of distribution of S. lyratus. The occurrences points overlap a 33% and the EGD a 48% with woody trees (closed to open (>15%) broadleaved evergreen or semi-deciduous forest (>5 m)), while the overlap with shrub (closed to open (>15%) (broadleaved or needle-leaved, evergreen or deciduous shrubland) was a 31% and 21%, respectively (Table 1).

Table 1 Percentages of overlap of the distribution model of Stephadiscus lyratus and occurrence points with land cover categories from Globcover 2009 dataset (UCLouvain & ESA Team).

Landcover	Overlapping (%)	
EGD	Occurrences	
Closed to open (>15%) broadleaved evergreen or semi-deciduous forest (>5 m)	48.67	33.33	
Closed to open (>15%) (broadleaved or needleleaved, evergreen or deciduous) shrubland (<5 m)	21.04	30.77	
Closed (>40%) broadleaved deciduous forest (>5 m)	5.77	0.00	
Closed to open (>15%) herbaceous vegetation (grassland, savannas or lichens/mosses)	4.41	12.82	
Mosaic cropland (50–70%)/vegetation (grassland/shrubland/forest) (20–50%)	2.66	0.00	
Mosaic vegetation (grassland/shrubland/forest) (50–70%)/cropland (20–50%)	2.02	5.13	
Rainfed croplands	1.95	0.00	
Open (15–40%) broadleaved deciduous forest/woodland (>5 m)	1.75	2.56	
Water bodies and Others	11.72	15.38	

The proportion of the potential distribution area in protected areas was 14.7%, occurring in 16 protected areas, from which six of them are located in Argentina while the remaining are from Chile. The higher proportional area protected is due to National Parks “Alberto D’Agostini” and “Yendegaia” (Chile), and Multiple Use Provincial Reserve “Corazon de la Isla Tierra del Fuego” (Argentina). Taking into account the category of management of IUCN, nine are category II National Parks and Ib Wilderness Nature Reserve; the six remaining are category IV Forest Reserve and VI Multiple Use Provincial Reserve (Table 2).

Table 2 Protected areas overlapped with distribution model of S. lyratus. Names, IUCN categories, country of origin and number of S. lyratus EGD pixels within protected areas.

Name of protected area	Design type	IUCN category	Country	Pixels (N°)	
Alberto D’Agostini	National Park	II	Chile	3,065	
Bernardo O’Higgins	National Park	II	Chile	56	
Cabo de Hornos	National Park	II	Chile	694	
Corazón de la Isla	Multiple Use Provincial Reserve	VI	Argentina	1,599	
Isla de los Estados y Archipiélago de Año Nuevo	Wilderness Nature Reserve	Ib	Argentina	703	
Kawésqar	National Park, National Reserve	II, IV	Chile	868	
Laguna Negra	Multiple Use Provincial Reserve	VI	Argentina	27	
Laguna Parrillar	Forest Reserve	IV	Chile	9	
Los Glaciares	National Park, National reserve and World Heritage Site	II, VI	Argentina	41	
Magallanes	Forest Reserve	IV	Chile	16	
Patagonia	National Park	II	Chile	11	
Rio Valdez	Multiple Use Provincial Reserve	VI	Argentina	57	
Seno Almirantazgo	Marine and Coastal Protected Area	IV	Chile	102	
Tierra del Fuego	National Park	II	Argentina	932	
Torres del Paine	National Park	II	Chile	836	
Yendegaia	National Park	II	Chile	1,687	

Discussion

Taxonomy and species morphology

Features of the gastropod shell have always been an essential and convenient source of taxonomic information. However, it is well recognized that shell characters such as shape, coiling patterns, and ribbing can be convergent and often mask crucial differences in anatomy (Stanisic, 1990; Barker, 2001). Hylton Scott, understanding the value of anatomical information, was the only researcher to provide anatomical descriptions of some South American charopids such as Stephacharopa testalba, Stephadiscus lyratus, S. mirabilis, Zilchogyra leptotera (Mabille, 1886). Recently, a study on Punctoidea phylogeny provided molecular information on some charopid species (Salvador et al., 2020). Stephadiscus lyratus, as all the species of the genus, has similar ornamentation in the shell protoconch and the teleoconch, without a marked limit between them. The type of ornamentation and similitude between protoconch and teleoconch differences Stephadiscus from all other South American charopid genera (Miquel & Cádiz Lorca, 2008; Miquel & Barker, 2009; Miquel & Araya, 2013). Inner anatomy information proves to be an essential source of characters relevant for future taxonomic and phylogenetic studies. The presence of a be-lobed kidney with the rectal side lobe bigger than the pericardial portion, plus the presence of a close secondary ureter, clearly indicates that S. lyratus belongs to Charopidae, differencing it from the Endodontidae. Solem (1982) raised these characters as the main differences between both families. The terminal portion of the uterus (spermoviduct), forming a compact glandular mass, is also a striking character that has only been mentioned before for Stephacharopa testalba. Along with this, the insertion of the vas deferens into the epiphallus through a dilatation constitutes unique structures of S. lyratus.

Estimation of the potential geographical distribution (EGD) and spatial analysis

The new records obtained were associated with native areas of the Magellanic Subpolar Forest sub-ecoregion in Argentina. The transferred model shows a potential distribution of S. lyratus to the Valdivian Temperate Forest, spreading the known area to a new sub-ecoregion, corresponding mainly to the area of this mentioned zone in Chile. Although no specimens of S. lyratus have been found in the Valdivian forest areas in Argentina (Nahuel Huapi and Los Alerces National Parks), these surveyed areas are outside to the east of the potential area predicted by the model. Thus, the potential area of distribution of S. lyratus is coincident with the Magellanic Subpolar Forest and the Valdivian Temperate Forest (more towards Chile) together with its boundaries with the Patagonian Steppe ecoregion. The obtained hypothesis of the potential distribution of S. lyratus will direct the next exploratory surveys with the expectation to find new populations in the future.

The biogeographic Valdivian Forest province according to Kuschel (1960) and Morrone (2018) has faunistic relationships with Magellanic Forests, probably because both regions have been isolated from other South American forests since the Neogene (Axelrod, Kalin Arroyo & Raven, 1991). In addition, the cooling cycles followed by warmer periods in the Quaternary caused the contractions and expansions of the temperate forests (Villagrán & Hinojosa, 1997), and some areas in the coastal range remained free of ices and may have been the source for the recovery of the forest biota (Smith, 2017). From the seven environmental variables used in the EGD analysis of S. lyratus, the main constraining variables are the temperature (BIO3 = Isothermality, BIO6 = Minimum temperature of coldest month, and BIO4 = Temperature seasonality) and Annual precipitation (BIO12).

Other studies using native snails in South America that analyzed the role of abiotic factors concerning their distribution are scarce e.g., the case of Megalobulimus sanctipauli (Ihering & Pilsbry, 1900). This kind of study for native micro snails is virtually nonexistent. Megalobulimus sanctipauli, known to inhabit the Atlantic Forest, showed temperature and rainfall as the determinant factors of their geographic distribution (Beltramino et al., 2015). In the case of Achatina fulica Bowdich, 1822, an exotic land gastropod in South America, Temperature seasonality, and Mean temperature of the coldest quarter were the variables that contribute the most to the model when they were used alone (Vogler et al., 2013). In the case of S. lyratus, the bell-shape response of environmental suitability for the BIO4, BIO6, and BIO12 show that the studied species try to avoid extreme temperature and precipitation oscillations. The BIO6 variable shows predicted suitable conditions at sub-zero temperatures. It suggests that this climatic variable is relevant for its optimal growing conditions and in particular with the species’ winter survival. Indeed, S. lyratus is found in the Magellanic Subpolar Forest where winter temperatures can drop below 0 °C. Empirical data on the life history of small land snail species in their natural habitat are hardly available worldwide. There is also very little data on gastropod cold hardiness, especially on land snail species of small sizes. Ansart & Vernon (2003) sustained that two alternatives exist for organisms living in areas that freeze in winter, such as the southern forest in Patagonia: move to an unfrozen habitat or face freezing conditions. For this last category, avoid freezing by extensive supercooling (freezing avoidance) or survive freezing of the body fluids (freezing tolerance) are the only possibilities. Freezing avoidance involves the choice of a hibernation site, which buffers the temperature differences, and which permits avoidance of inoculative freezing (e.g., by contact with ice). During hibernation, the snails rapidly suppress their metabolism and minimize water loss using a discontinuous gas exchange pattern (Koštál et al., 2013). We sustain that S. lyratus find favorable overwintering microhabitat in soil, under tree bark, or under fallen tree trunks in contact with soil, which is well buffered from temperature and moisture fluctuations allowing the species to survive during winter freezing.

Decaying wood can absorb and retain water for several weeks during periods of low precipitation, providing a buffer from microclimate extremes (Jordan & Hoffman Black, 2012). Land gastropods require moisture for respiration and locomotion, and humid microenvironments are known to be a prerequisite for the occurrence of many land mollusk species. However, excessive wet extremes that lead to flooded soils for long periods are not tolerated by most micro snail species buried in the soil (Addison & Barber, 1997). This can be a restrictive situation to Stephadiscus lyratus occurrence as it is shown by the predicted environmental suitability in a narrow range of annual precipitation not surpassing around 1,500 mm per year. Also, this would explain the occurrence of S. lyratus in humid areas of Magellanic subpolar forest, and the projected potential environments in some regions of northern Valdivian Temperate Forest, which featured by 1,000 mm of precipitation per year (https://www.worldwildlife.org/ecoregions/nt0404).

Stephadiscus lyratus, a vulnerable species to protect

Stephadiscus lyratus inhabit mostly woody areas of native forests. The preferred microhabitats of the species are sites on or under the bark of fallen trees or under humid logs in contact with the soil. These microhabitats are less frequent in disturbed forest areas with high human pressure located in Tierra del Fuego outside the National Park, where specimens were difficult to find alive. In addition to microclimate effects, coarse woody debris has a significant influence on gastropod food availability. Dead and decaying wood promotes a diversity of fungi, an important food resource for many snails. Land gastropods are suffering habitat loss and competition from introduced species (more numerous and prolific every year), although are regarded as non-charismatic groups for conservation purposes (Régnier, Fontaine & Bouchet, 2009; Régnier et al., 2015). In the case of Orthalicoidean land snail in Argentina, only 3% of their average species distribution ranges are safeguarded within current protected areas, showing that the existing protected areas system is not effective at all for the protection of this invertebrate group (Ovando et al., 2019). In the present study, we found that 14.5% of the total current distributional area of 72,672 km2 of S. lyratus is inside the system of protected areas. However, it is important to emphasize that the threats and pressures of land-use change such as tourism, logging, and frequent fires compromise both, protected areas inside the southern National Parks, and the matrix that surrounds them. In fact, 700–1,000 ha are logged each year (from 1980 to 2003) in Argentina, Tierra del Fuego (Gea-Izquierdo et al., 2004). Between 53.1% and 68.1% of the Chilean Magellanic forests are influenced by human activity in some way (Inostroza, Zasada & König, 2016). In this context, habitat loss for land gastropod conservation is very worrying. Moreover, the protected areas in Argentina were created for the protection of plants (Ortega-Baes et al., 2012) and/or vertebrates (Tabeni, Bender & Ojeda, 2004; Arzamendia & Giraudo, 2004; Corbalán et al., 2011; Tognelli et al., 2011), ignoring the invertebrates, even when their importance has been proven (Chehébar et al., 2013). This situation is a worldwide problem since there are more than a million invertebrate described species but only 3,500 are protected (Baillie, Hilton-Taylor & Stuart, 2004; Brooks et al., 2004, 2006; Nieto et al., 2017). We sustain that S. lyratus find favorable overwintering microhabitat in soil or under stones in contact with soil, which is well buffered from temperature and moisture fluctuations allowing them to survive during winter freezing. The combination of the ectothermic traits of this species, low dispersal capacity, probable low fecundity producing only a few eggs, and its narrow habitat requirements (forest specialist) turns S. lyratus into a potentially vulnerable species. Along with habitat loss through human land use, climate change is a major contributor to biodiversity loss in the 21st century (Lee et al., 2015). The climate is changing rapidly, and terrestrial ectotherms are expected to be particularly vulnerable to an increase in extreme weather events in temperate regions (Nicolai & Ansart, 2017). They will be affected seasonally by more frequent hot temperature extremes and fewer cold temperature extremes over most land areas (IPCC, 2014). Meanwhile, the projected precipitation changes show reductions for the dry area in the central-western region and the whole of Patagonia (Barros et al., 2015). In the Magellanean region, there have been reported extreme events that could be increased in frequency and intensity in the context of climate change. Examples of these events were severe droughts (1920–1926, 1928, 1966), heavy rainfall events with floods (1983, 1990, 2012, 2015), and devastating snowfall storms. Thus, the quality and availability of habitat for S. lyratus could be compromised by the effect of both land uses changes and global warming. In this context, these regional changes in climate and land use put S. lyratus populations at serious risk of extinction that must be taken into consideration for future conservation actions.

Conclusions

Here we confirm that the shell ultrastructure of S. lyratus has a protoconch and teleoconch with similar ornamentation, not showing a marked limit between them. We provide new anatomical information highlighting the presence of a be-lobed kidney, a close secondary ureter, and the terminal portion of the uterus (spermoviduct) forming a compact glandular mass, the vas deferens inserting into the epiphallus through a dilatation as the most notable anatomical characters of S. lyratus.

The potential distribution obtained shows that S. lyratus could be found beyond the Magellanic Subpolar Forests into the Valdivian Temperate Forest, which would increase its known distribution area to a new sub-ecoregion, mainly within Chile.

From seven environmental abiotic variables used, the main constraining ones to explain S. lyratus occurrence in the EGD are the temperature (Isothermality, Minimum temperature of coldest month, and Temperature seasonality) and annual precipitation.

Stephadiscus lyratus inhabits cold native forest areas where it is found mainly on or under the bark of fallen trees or damp trunks in contact with the ground. This microhabitat allows them overwintering, buffered from temperature and moisture fluctuations, and survive during winter, probably as a strategy to avoid freezing.

The combination of its narrow habitat requirements (forest specialist), the ectothermic traits of this species, and its low dispersal capacity turn S. lyratus into a potentially vulnerable species, to current land-use change, and future climate change scenarios.

Supplemental Information

Supplemental Information 1 Occurrence records of Stephadiscus lyratus from field surveys, museum collections, and scientific articles.

Click here for additional data file.

Supplemental Information 2 Comparison of Stephadiscus lyratus with other close related South American species.

The gray column shows the low number of specimens on which the original descriptions were based.

Click here for additional data file.

Supplemental Information 3 Model performance of candidate models according regularization multiplier (RM) and feature classes (FC, l: linear, q: quadratic, p: product). The selection criteria were partial ROC (pROC), omission rate and AIC.

Click here for additional data file.

Supplemental Information 4 Settings and model performance of the final model. RM (regularization multiplier), FC (feature classes, l: linear, q: quadratic, p: product), pROC (partial ROC), omission rate and AIC.

Click here for additional data file.

Supplemental Information 5 MOP analysis for environmental conditions based on the training region “M” and the transfer area “G”.

Areas with the most dissimilar variables conditions (i.e., where one or more environmental variables are outside the range present in the training data) are represented by zero value. These areas represents strict extrapolative areas so predictions in those areas should be treated with strong caution. Other values represent levels of similarity between the calibration area and the “G” transfer area.

Click here for additional data file.

Supplemental Information 6 Species discovery .

Click here for additional data file.

Supplemental Information 7 Script used for the analyses in "R".

Click here for additional data file.

We would like to thank the curators and researchers from the different collections and databases consulted, Alejandro Tablado and Sergio Miquel, Museo Bernardino Rivadavia (MACN-In); Gustavo Darrigran, Museo de La Plata (MLP); Ellen Strong, Smithsonian National Museum of Natural History (NMNH); Adam Baldinger, Museum of Comparative Zoology (MCZ), and the Academy of Natural Sciences in Philadelphia (ANSP). To the National Parks agency of Argentina for providing permits to work in Tierra del Fuego, Los Alerces, and Nahuel Huapi National Parks. A special thanks to Los Alerces park rangers, Florencia Pantasso, and Dario Bassoso, who kindly support and facilitate all fieldwork. MGC would like to thank the Centro Austral de Investigaciones Científicas (CADIC-CONICET) for allowing us to use their facilities during December 2018 and January 2019 for this research. A special thanks to Christopher Anderson, who kindly offered space in his laboratory and helped to obtain all permits to work in Tierra del Fuego. To Marina Tagliaferro for her kindness and support in the CADIC microscopy laboratory. To Christopher Anderson and Alejandro Valenzuela for their kind hospitality while in Ushuaia and to Los Dominguez that helped with the fieldwork. To A. Lira-Noriega and L. Osorio-Olvera for teaching and supporting us with the methodology and use of the Kuenm package for R. We thanks the Editor, JJ. Morrone, and the reviewers, E. Martínez-Meyer, T. Escalante, and an anonymous reviewer for providing valuable comments on the manuscript.

Additional Information and Declarations

Competing Interests

Author Contributions

Field Study Permissions

Data Availability

The authors declare that they have no competing interests.

Maria Gabriela Cuezzo conceived and designed the experiments, performed the experiments, analyzed the data, prepared figures and/or tables, authored or reviewed drafts of the paper, and approved the final draft.

Regina Gabriela Medina conceived and designed the experiments, performed the experiments, analyzed the data, prepared figures and/or tables, authored or reviewed drafts of the paper, and approved the final draft.

Carolina Nieto conceived and designed the experiments, performed the experiments, analyzed the data, prepared figures and/or tables, authored or reviewed drafts of the paper, and approved the final draft.

The following information was supplied relating to field study approvals (i.e., approving body and any reference numbers):

Specimens collection were approved by the Argentine National Parks Administration and carried out in: National Parks Tierra del Fuego (DRPA 146/2018-19), National Parks Nahuel Huapi and Los Alerces (DRPA 1674/2019). Also within areas of Chubut province permisions were approved by Direccion de Fauna Silvestre de la Provincia de Chubut (DFyFS1/19).

The following information was supplied regarding data availability:

Occurrence records of Stephadiscus lyratus used in this work and obtained from field surveys, museum collections, and scientific articles are available in Table S1.

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
