# Peer review of "Geographic distribution modeling and taxonomy of Stephadiscus lyratus (Cothouny in Gould, 1846) (Charopidae) reveal potential distributional areas of the species along the Patagonian Forests"

_PeerJ, doi:10.7717/peerj.11614_

## Round 0.1 · original submission · Major Revisions

I have received three reviews, one of them suggests that the paper is rejected and another major revision. I would be willing to consider a revised version of the manuscript, that especially takes into account the concerns of these reviewers.

Reviewer 1 ·

Basic reporting

The authors need to improve their English in some parts (see specific comments). It remains to improve the information in the introduction, includes more examples of how rare the study of micro-snails; provide microhabitat data and its importance for the persistence of snails in similar latitudes, temperate ecosystems, or neighboring countries such as Chile. PeerJ standards are fine. Figures can be better, especially Figures 1 and 5. Raw data is fine.

Experimental design

The authors should improve some aspects of the transects and provide more details to ensure replicability. The niche distribution model is well explained, but you should provide examples of similar work with smaller numbers of samples for reliable models. The questions could be more precise.

Validity of the findings

The new antecedents for this species are relevant and must be considered and validated, but improvements are required. The new morphological data of the shell and the internal parts are quite good. You need yes or yes to improve the microhabitat part. Improve the delivery of information in the niche model.

Additional comments

Thank you for giving me the opportunity to review the work of the authors Cuezzo, Medina & Nieto. The work consists of showing new morphological evidence of both the ultrastructure of the shell and the internal parts of Stephadiscus lyratus, a forest micro-snail. Furthermore, due to the scarce information on these snails, the authors provide data on microhabitats, a potential model of an ecological niche, and the representation of this species of micro snail in the system of protected areas of the states of Chile and Argentina. The work is a good contribution from the taxonomic and ecological point of view. I agree that the authors are making an important contribution to a species of which almost nothing is known. However, better work needs to be done. In the current form, the authors need to improve the background in the introduction, clarify their objectives and some methods. In addition, the results present necessary information that must be removed since it does not contribute to the development of the proposed objectives. The authors are not conducting a systematic review of the genus Stephadiscus, but rather providing new morphological and ecological data, which I repeat is tremendously valuable. In the discussion, the authors focus more on their results, but they speculate on some points and that should be corrected.

Introduction
Line 2. Remove this: “Stephadiscus Hylton Scott, 1981 is a genus of the diverse…” Star with “Charopidae is…”
Line 6. Remove: “was the researcher that most..” and after described insert “ many”
Line 8: please specify what you mean by anatomical (shell morphology, internal parts morphology?).
Line 9: this phrase “Also, adequate fieldwork to estimate the current distributional range of genera and species has not been done” is an affirmative sentence. Are you sure that any other researchers made this task before?
Line 11. Insert here: Stephadiscus, Hylton Scott, 1981, included…; delete “initially”
line 17. Add “Currently, the distribution of Stephadiscus…”
Lines 15-17. “The taxonomic position of Stephadiscus striatus Hylton Scott, 1981 from northeast Argentina and Venezuela, will have to be reconsidered since it seems to belong to Punctidae (Miquel & Barker, 2009)” please rephrase. It can read some rare this phrase.
Line 18. Change: “The distribution of the Stephadiscus” to “The Stephadiscus distribution is restricted to…”
Lines 21-24. This phrase is some unconnected from the main text “Biogeographically, southern South America belongs to the Andean region, proposed among others by Morrone (2018), who divided this region into three subregions, the Subantarctic, Central Chilean and Patagonian. The Andean region has a closer relationship to the Austral region in the Austral kingdom (Morrone, 2018)” I suggest change the sense and combine phrases, for example, after “The distribution of the Stephadiscus is restricted to Patagonia at both sides of the Andes from S 36° towards the southernmost portion of the continent, including Malvinas islands and southern archipelagos (Miquel & Barker, 2009; Miquel & Araya, 2013) and belong to the Andean region following Morrone (2018). This region was divided into three region: Subantarctic, central Chile and Patagonian (Morrone 2018?)
Line 25. Delete from this line this: “designated as the genus type species” and incorporated in brackets in line 13 after lyratus
Line 28. This have some cite or is only a personal observations of the authors?
Line 37. Please provide other cites. There is a wide literature about modelling
Line 38. Delete “the” species…
Line 39. Delete, in this sense and rephrase please.
General comment: is important for this work reveal that modelling niche for land snails is an important tool, especially to know your distribution/potential distribution. In current form, the introduction needs more information. I suggest read, and incorporate these articles:
1. Weaver, K. F., Anderson, T., & Guralnick, R. (2006). Combining phylogenetic and ecological niche modeling approaches to determine distribution and historical biogeography of Black Hills mountain snails (Oreohelicidae). Diversity and Distributions, 12(6), 756-766.
2. Beltramino, A. A., Vogler, R. E., Gregoric, D. E. G., & Rumi, A. (2015). Impact of climate change on the distribution of a giant land snail from South America: predicting future trends for setting conservation priorities on native malacofauna. Climatic Change, 131(4), 621-633.
3. Mumladze, L. (2014). Sympatry without co-occurrence: exploring the pattern of distribution of two Helix species in Georgia using an ecological niche modelling approach. Journal of Molluscan Studies, 80(3), 249-255.
General comment: the authors show a second aim: “second, to characterize the microhabitat where it is found”. However, nothing about microhabitat is said in the introduction. Microhabitat can influence distribution and abundance of land snail by environmental temperature or food supply and is essential for the occurrence of them. Also, obviously, can influence the potential distribution using modelling. As the ecology of land snail in south America is scarce, I suggest taking account a new paragraph with articles in similar latitudes or ecosystems as:
1. Barahona-Segovia, R. M., Riveros-Díaz, A. L., Zaror, S., Catalán, R., & Araya, J. F. (2019). Shelter, ecophysiology and conservation status of Plectostylus araucanus (Pulmonata: Bothriembryontidae) in the fragmented Maulino Forest, central Chile. Revista Mexicana De Biodiversidad, 90, 1-11. (For Chile)
2. Gómez, P., Espinoza, S., Hahn, S., Valenzuela, M., & Ormazábal, Y. (2020). Forest variables associated to the occurrence of the endemic pulmonate Macrocyclis peruvianus (black snail) from the Maule Region of central Chile. Caldasia, 42(2), 306-312. (For Chile)
3. Murray, J., Johnson, M. S., & Clarke, B. (1982). Microhabitat differences among genetically similar species of Partula. Evolution, 316-325. (For Micronesia)
4. Müller, J., Strätz, C., & Hothorn, T. (2005). Habitat factors for land snails in European beech forests with a special focus on coarse woody debris. European Journal of Forest Research, 124(3), 233-242. (For Germany)

M&M
Lines 63-65. Cite? I suggest Smith-Ramírez, C. (2004). The Chilean coastal range: a vanishing center of biodiversity and endemism in South American temperate rainforests. Biodiversity & Conservation, 13(2), 373-393.
Line 70. Jungle? This is wrong. Valdivian evergreen forest is a rainfall forest but not a jungle. Remove this please.
Line 78. Please provide cites.
Line 95: how meters have you linear transect?
Lines 96-99. How many roots, trunks or stones survey per transect? What is the sampling effort per site? How many persons per site? More details are necessary to ensure replicability.
Lines 128-129. Delete “as part of the 60”
Line 141. Are 24 points a good number? When I work in modeling, the minimum required is 30 or 40 records for the most reliable distribution model. Please provide other articles when after spatial autocorrelation delete many occurrences.
Line 171: full occurrences are 60, 37 or 24? Please specify. If are 24, please remember to give more examples with a low number of occurrences. If are 60, why use full dataset if you are applied a spatial correlation?
Line 171. Final models or final model? Please be consistent through the text
Line 75. Define G region here

Results
Lines 201-220. Your work is not a systematic review, therefore these phrases are unnecessary here. Please remove.
Line 223. Delete body
Line 224. Add “body” black…
Lines 224-260. Why the authors combine taxonomy with ecological modeling? This link in the introduction is missing or at least, poorly developed.
Line 264. Stephadiscus lyratus was mainly found living on or under the bark of fallen trees or under humid logs in contact with the ground. How is percentage per microhabitat? The authors realize transect and on each, search actively land snails per microhabitat type, therefore, authors can obtain a percentage of microhabitat use.
Line 267. Rephrase
Line 269. Temperature buffer? Please clarify
Lines 275-334. This is completely unconnected with the aims proposed by the authors. In a taxonomical article this information must be placed in the introduction in many cases, for example, “S. lyratus has the largest shell diameter (DM= 5.5 mm) among all species classified in Stephadiscus. It is a very conspicuous species regarding its shell coloration and sculpture”, can include in the present work in introduction section. Again, I think that this a hybrid work with Taxonomy and modeling niche and this link must be clear. Remove this lines of your present work
Line 340. Met? Also, show the AICc value for best fitted model and compare it with other models.
Lines 342-345. What is the values of AICc weight for BIO variables?
Line 367. Which IUCN category are II, Ib respectively? Please rephrase

Discussion
Line 378. Deleting the information of lines 275-334, you must place this authority names in this line-
Line 380. Always, after a continuum point, you must provide the full name of the species. Review through the text.
Lines 382-383. Please rephrase.
Line 398. To the area in Chile? Replace for “to Chile”
Lines 411-415. “From the seven environmental variables used in the EGD analysis of S. lyratus, the main constraining variables are the temperature (Isothermality, Minimum Temperature of Coldest Month, and Seasonality) and Annual Precipitation. Thus, temperature and precipitation were the most important abiotic factors to explain S. lyratus occurrence”
What do you hope will happen to the micro snail with climate change? How will the abundance or distribution of the micro snail vary with potential changes in humidity and temperature? Please include a short text about it.
Lines 415-420. Sentence too long, please separate into two phrases. In addition, use this example in the introduction to show the scarce sources of information in this taxonomic group.
Lines 423-427. This part begins with the importance of microhabitats for S. lyratus, but in the introduction, the authors not present anything of this structural key element. Moreover, the description is very qualitative and the authors not present percentages, proportions or other metrics to reveal the importance of fallen barks, for example, we find 60% of all individuals under fallen barks vs 12% of individuals in logs in contact with humid soil. This type of information is key to understand future conservation status if the aim of this part is that.
Lines 428-430. Provides cites. I suggest:
Régnier, C., Achaz, G., Lambert, A., Cowie, R. H., Bouchet, P., & Fontaine, B. (2015). Mass extinction in poorly known taxa. Proceeding of Natural Academy of Sciences, 112, 7761–7766.
Régnier, C., Fontaine, B., & Bouchet, P. (2009). Not knowing, not recording, not listing: numerous unnoticed mollusk extinctions. Conservation Biology, 23, 1214–1221.
Line 448. Change “evaluation” for “assess”
Line 458. Small size?
Lines 468-471. Provide cites
Lines 471. In line 455, the authors said that “Empirical data on the life history of small land snail species in their natural habitat are hardly available worldwide” but, in the 471-473, the authors said “The combination of the ectothermic physiological traits of this species, low dispersal capacity, probable low fecundity producing only few eggs, and its narrow habitat requirement (forest specialist) turns S. lyratus into a vulnerable species”. The authors realized experimental thermal or supercooling experiment in S lyratus? Do the authors present information on dispersal capacity or habitat requirements? I understand that is probable, but, the information is not available and therefore, you can suggest that this species could be vulnerable, but not use affirmative sentence.
If you want to declare S. lyratus as vulnerable species must apply an IUCN Red List, for example, B criterion, the most rapid assessment for this species and see if it meet the thresholds
Lines 457-471. The authors explain largely physiological traits that could be the cause for S. lyratus to be considered a vulnerable species, but, is too long explanation for a final speculation. Reduce importantly this and be concise.

Conclusions
Line 497. Anatomical parts is internal parts? Review through the text
Point 5 of conclusion is not supported and is only speculation without experimental or natural dataset and therefore I suggest remove it. Conclusions must be of your results.
In addition, one aim is to characterize the microhabitat, but the authors nothing concludes about this. Please replace the point 5 with the current information by microhabitat conclusion

Figures
Figure 1. The green color of Patagonian forest is to clear
Figure 5. Describe what mean VTF, MSF, etc… the legend must be sufficient to understand all aspect of the figure

Annotated reviews are not available for download in order to protect the identity of reviewers who chose to remain anonymous.

·

Basic reporting

All my comments are in the General comments section.

Experimental design

All my comments are in the General comments section.

Validity of the findings

All my comments are in the General comments section.

Additional comments

General comments
In this contribution, the authors present new data on the morphology and distribution of the micro snail Stephadiscus lyratus in the Patagonian Forests. Also, they developed a niche model to identify potential distribution areas in neighboring subecoregions. Their results showed new morphological diagnostic traits for the species and that suitable climates exist in the Valdivian Temperate Forests subecoregion.

I am not an expert in snails, so there is nothing that I can comment on the taxonomic findings. Still, I realize that these contribute to the biology and taxonomy of the group due to the authors' careful morphological analyses that, as they mentioned in the manuscript, are not common for these invertebrates.

I will focus on the distributional data and analysis. I think that the niche modeling was technically well conducted. The authors were strict in the methodology and presented all the information necessary to replicate the analysis. My concerns, however, are in the motivation of this analysis and the interpretation of their results.

First, If the idea was to test the hypothesis that this snail has a broader geographic distribution than the one proposed by Scott (restricted to the Subpolar Magellanic Forest), I think that the work is incomplete. This is because with the niche model you cannot test this hypothesis; actually, the potential distribution obtained with the niche model is a hypothesis itself, so you need field data from the region where the model indicated that there are suitable conditions for the species to test the initial hypothesis.

Second, although the modeling procedure was technically adequate, its use was overly simplistic. The authors used it only to describe the potential distribution of the species and the variables that were more informative according to maxent, without any attempt to understand why those variables were important and the implications of this information in the light of conservation, climate change and other topics that the authors present in the Discussion. In other words, this contribution is mostly descriptive but very little analytic. I strongly suggest that the authors make an effort to make a stronger link between the morphology that they studied here, the physiology known for the species and the distribution, to present more robust support for their inferences.

Specific comments
L34. New paragraph for Modern methodologies…

L36. Eliminate and/spatial. ENMs use only the environmental data obtained from the locations.

L43-45. I understand what you mean, but it is not correctly expressed. It is not by identifying scenopoetic variables related to the species that makes it possible to identify areas where other populations inhabit. It is the combination of environmental conditions in the relevant scenopoetic variables that really matter.

L54. There is no mention of collecting permits.

L61. We followed…

L73. Not vegetation, but plants

L158. Actually, several of your environmental variables are correlated, but not at the level of your threshold, which, by the way, is pretty high.

L275. I think this section, while interesting, is overly long. I suggest sending it all or at least most of it to Supplementary Information as it does not make part of the findings of the study. Also, it should not go in the Results section, but in the Discussion.

L411. New paragraph for From the seven…

L414-415. You only used temperature and precipitation information to build the models, so this sentence is tautological. Here it would be useful to discuss why such environmental variables were the important ones for the snail distribution. In other words, what limits the distribution of the species and why.

L.475-492. This part of the Discussion regarding the possible effects of climate change in the snail can be improved with actual data specific for the region. Not only future projections but also if there is something that the authors can say about how the climate has changed in Magellanic Subpolar Forests in the last decades and how these changes affect the species' physiology in light of your findings. For example, regarding the environmental variables that you found important for the distribution of the species. Otherwise, it is just speculation

L501-505. This conclusion is definitively misleading. The fact that the niche model indicated that there are suitable climatic conditions for the snail in the Valdivian Temperate Forest does not mean that the species is there, and thus that your initial hypothesis is not supported. You obtained a distributional hypothesis with your modeling analysis, not a test of your initial one; you cannot test one hypothesis with another hypothesis. Moreover, your fieldwork found no evidence of the species' presence in the area of the Valdivian Temperate Forest where you sampled. As you mentioned, your results may guide future fieldwork in the areas where the model indicates potentially suitable areas, but until then, you cannot accept or reject your initial hypothesis.

Figure 5. It would be helpful to present in the map the occurrence and sampling sites (where you did not find the species). Also, you can include the protected areas that you used in the analysis.

Table 1. I think that If you are interested in an analysis of habitat preference, it is better to do it with the occurrences, not with the potential distribution map.

·

Basic reporting

No comment

Experimental design

No comment

Validity of the findings

No comment

Additional comments

The manuscript is adequate in content, methodology and results.

---

## Round 0.2 · Minor Revisions

I find that the reviewer is making some valid points that should be addressed in a revised version. Please read the review thoroughly and make the suggested changes, so that I can accept the manuscript.

Reviewer 1 ·

Basic reporting

1. English editing service is necessary through the text. I suggest an excellent tool for improving this problem in the review letter.
2. Literature reference provided sufficient background.
3. Article adequately build.
4. Self-contained with relevant results to hypotheses.

Experimental design

1. Follow primary research, although some results still need to connect with the aims proposed.
2. Research question well addressed.
3. High technical and ethical standards.
4. Methods well developed.

Validity of the findings

1. For micro snails, is important to understand more about their ecology.
2. For modelling is well supported
3. Bullet point in the conclusion needs work and connected with the original aim.
4. Some speculation is provided, although not all well supported.

Additional comments

The manuscript improves your content, but English still needs review through the text. I suggest an excellent tool for non-native speakers called trinka.ai (https://www.trinka.ai/). This an efficient Grammarly and writing English corrector used specifically for non-native speakers and scientists in many parts.

Also, I do not agree with some parts of the answers given in the rebuttal letter in the remarks section, since those comparisons can be made in a dichotomous species key, thus making sense of the previously mentioned systematics for this species. The systematic section needs improving.

Other comments as follow:

Title
¿Why the taxonomy reveals potential distribution areas? I suggest ordering the words used in the title.

Abstract
English review is needed.

Introduction
L80. “(Couthouy in Gould, 1846)”.
When the authors insert taxonomical description by some author included in the work of a second taxonomist, please use italic for “in”. The correct form according to the Zoological code of nomenclature and ZooBank is (Couthouy in Gould, 1846)”. Correct through the text

L111-112. Add references for this affirmation.

L198. Delete “plus” and replace it with a ‘+’ symbol.

M & M
L214. What is the terminology used by the authors? This is key in the description of the internal anatomical parts. Please provide the most updated terminology used in malacology.

Results
L388. “Rainforest provide…” or temperate forest? be consistent. I think that the most adequate term is temperate.

L398-399. “but probably are associated…” This is speculative and I suggest removing it. However, the authors can discuss this briefly.

L411. After the dot place the complete name of the genus “… 1.5mm). Stephadiscus lyratus …”

L414. “Couthouy in Goul” in italic

L407-485. I do not agree with the author putting the comparisons with other Stephadiscus species and other genera in full text ¿Why don't the authors incorporate a dichotomous species key using whorls, shell, and other taxonomical traits and solve the systematics from this point of view? This resolves my doubts about why systematic and not a simple taxonomical ordination. In addition, I think that the synonyms, the type species, and syntype for your work do not add new information for the species because this already described. Also, I think is necessary to delete lines 328-341. A systematic review is not proposed in your aims, so, are necessary changes about this theme.

L499-503. Based on the proposals made by the second reviewer, the authors incorporate in lines 298-315 a full paragraph for the answer. However, I find it unnecessary, in results, to describe what the BIO3 means (499-503).

Discussion
L556-558. For this reason, the dichotomic species key is completely necessary. The differences with other genus or even, with similar species of the same genus might be resolved using this tool.

L559. Internal anatomical parts?

L587. Change to the format of reference citations in text.

L593. “… main constraining variables are (related with) the temperature…” many grammatical errors like this are present in the text. Please, use trinka.ai or a specialized English edition service.

L605. “Temperature Seasonality, and Mean Temperature” not use a capital letters.

L607. ¿Bell shaped or bell-shaped like the results? Be consistent.

L608-609. Not use a capital letters for BIO variable names.

L637. References?

L652. Add a dot at the end.

L663. References?

L667-668. Use format for references in the text.

L679-682. I think that this paragraph is unconnected from the conservation issues and is repeated literally from the lines 626-629. Please, authors must be careful with this.

L683-685. I still considered that this is speculation of the authors and not based on the information available. However, is true that land snails have low vagility and that they are vulnerable to habitat fragmentation, fires, or climatic change, but not, for this reason, the statement in the rebuttal letter “does not deserve any additional experimentation to sustain it” is true.

Conclusions
Bullet 5. The statement “The combination of the ectothermic physiological traits of this species, low dispersal capacity…” not supported by the current data. When authors say physiological trait, I imagine, e.g., rollover speed, supercooling or freezing point, or physiological performance at different controlled temperatures to permit the readers to understand which are the environmental limits for the species. These limits are not the environmental variables explored here because, as the authors mentioned in your manuscript, microhabitat buffer these environmental conditions, and therefore, the performance can change under these conditions. Also, the physiological traits can change along latitudinal gradient permitting several phenotypical expressions for the same environmental pressure.

Acknowledgments
Include editor and reviewers.

Annotated reviews are not available for download in order to protect the identity of reviewers who chose to remain anonymous.

---

## Round 0.3 · accepted · Accept

Thanks for providing a revised version of your manuscript!